# Employing a MEMS plasma switch for conditioning high-voltage kinetic energy harvesters

Hemin Zhang [1,2], Frédéric Marty[1], Xin Xia[3], Yunlong Zi [3], Tarik Bourouina [1], Dimitri Galayko [4✉] & Philippe Basset [1✉]

Triboelectric nanogenerators have attracted wide attention due to their promising capabilities of scavenging the ambient environmental mechanical energy. However, efficient energy management of the generated high-voltage for practical low-voltage applications is still under investigation. Autonomous switches are key elements for improving the harvested energy per mechanical cycle, but they are complicated to implement at such voltages higher than several hundreds of volts. This paper proposes a self-sustained and automatic hysteresis plasma switch made from silicon micromachining, and implemented in a two-stage efficient conditioning circuit for powering low-voltage devices using triboelectric nanogenerators. The hysteresis of this microelectromechanical switch is controllable by topological design and the actuation of the switch combines the principles of micro-discharge and electrostatic pulling, without the need of any power-consuming control electronic circuits. The experimental results indicate that the energy harvesting efficiency is improved by two orders of magnitude compared to the conventional full-wave rectifying circuit.

[1] ESYCOM, Univ Gustave Eiffel, CNRS, CNAM, ESIEE Paris, F-77454 Marne-la-Vallée, France. [2] Department of Engineering, The Nanoscience Centre, University of Cambridge, Cambridge CB3 0FF, UK. [3] The Chinese University of Hong Kong, Shatin, N.T., Hong Kong SAR, China. [4] Sorbonne Université, LIP6, Paris, France. ✉email: dimitri.galayko@sorbonne-universite.fr; philippe.basset@esiee.fr

There is a great demand for micro-to-milli watts power sources with the rapid development of implantable devices and wireless sensing nodes[1]. Compared to chemical batteries[2,3], the energy harvesters that transduce the broadly existing environmental energy to electricity have the advantages of renewability, flexibility and sustainability[4]. Among kinetic energy harvesters using electrostatic electret[5–8], piezoelectricity[9–11], electromagnetism[12,13] or triboelectrification[14–19], triboelectric nanogenerators (TENGs) are promising solutions. TENGs are basically electrostatic kinetic energy harvesters (e-KEH)[20]. They can be seen as capacitive transducers providing their own electrical biasing mechanism due to the contact electrification effect[21]. First works on e-KEH considered "bare" variable capacitors which were supposed to be pre-biased by external sources, so that these devices were not self-sufficient[22]. The next generation of e-KEH employed an electret layer for initial biasing which contained embedded non-compensated electrical charges, however, the depolarization with time of the electret became the main shortcoming. Triboelectric effect provides a similar biasing solution as an electret layer, with a notable difference: the embedded charges in the dielectric layer are (re)generated during the device operation[23–26].

The conditioning circuits for TENGs are still under development, as TENGs normally generate ultra-high voltage impulses with peaks largely above 100 V but low current with peaks of several µA to sub-mA[27]. To store the generated electrical energy in order to power low-voltage electronics, a specific conditioning circuit should include an AC-to-DC converter and a DC stabilization module. Basic conditioning strategies are based on full-wave (FW) or half-wave (HW) diode-bridge rectifiers, which charge a large capacitor to a DC voltage. However, such a strategy has two major drawbacks. First, the output voltage of the rectifiers maximizing the converted energy per cycle must be of the same order of magnitude as the embedded triboelectret biasing voltage $V_{TE}$ of a TENG (typically $V_{TE}/2$ for the FW bridge[20]), which is not compatible with the load supply. Second, it exists a maximum converted energy per cycle that is defined by (and increases with) $V_{TE}$[28,29], so that one may want to maximize this voltage; however, the latter is fundamentally defined by the triboelectric effect. This maximum in the conversion efficiency is explained by the saturation phenomenon of the FW/HW diode-bridges and other stable charge pumps like Dickson[30] or Cockroft–Walton[31–35] charge pumps.

Adding a synchronous switch before or after the rectifiers is a solution for improving the conversion efficiency. Motion-triggered mechanical switches[34,35], electrical switches[36–38] combined with their control circuits, or electrostatic switches[39–41] have been investigated. In these solutions, the switches need to be activated at each actuation cycle, thus requiring a precise synchronization with a specific value of the TENG capacitance, typically its maximum or minimum value. For the electronic switches, their actuation control was obtained thanks to a circuit powered by an external battery, at least for the start-up, making the system not self-sustained.

In order to optimize the interface with the load, 2-stage solutions[42] are often used. The TENG charges a small buffer capacitor to quickly reach a high optimal voltage which maximizes the conversion efficiency, and then a DC–DC converter transfers the energy from the buffer to a high-capacitance/low-voltage storage capacitor. Such configuration solves two problems. First, it allows to set a high voltage for the buffer capacitor so to maximize the energy transfer rate, while setting a low voltage at the storage capacitor to supply the load. Second, a small value of the buffer capacitor allows a quick circuit set-up[43]. However, it requires an external control for the switch that is quite power-consuming as aforesaid. Besides, additional energy dissipation will be brought in because solid-state electronic switches are inherently leaky. Superior electrical insulation between the two stages is mandatory, therefore an acceptable switch must have good physical disconnection properties when the switch is OFF. Furthermore, the DC–DC converter operates with a high duty cycle[39,42], i.e., the voltage across the buffer oscillates between a maximum voltage and the ground, which is particularly inefficient because of the wasted time for the buffer to recover the high-voltages from zero once the switch is actuated. To overcome these issues, a self-actuation switch with high actuation-voltage and narrow hysteresis loop is needed. Implementing such a switch with electronic circuits is more difficult and consuming because the switch control needs to involve two high-voltages thresholds instead of one.

In this paper, we report a self-sustained conditioning system that allows the TENG to work at high-voltages for high-energy conversion without power-consuming electronics, using an unstable charge pump (Bennet doubler) combined with a high-voltage microelectromechanical system (MEMS) plasma switch in a 2-stage circuit. The Bennet doubler solves the $V_{TE}$ voltage limitation issue by generating an exponential transient charging process with positive base of exponentiation[32,44,45]. Such a transient process has no saturation limit, except those imposed by the breakdown voltage of the device insulators[46], the possible spring-softening phenomena occurring in resonating devices[32] or limitations from electrical components. A high-voltage MEMS plasma switch is developed to control a buck converter transferring the energy between the buffer and the final reservoir. The switch control law is provided with an automatic narrow hysteresis loop, in order to hold the voltage across the buffer capacitor always oscillating between two high voltage levels. Compared to electronic switches, the proposed micro-plasma switch has the advantages of no electronic control, no ohmic contact and no need to be supplied with external energy. The proposed switch is a fully "stand-alone" device and does not require direct integration with the TENG. The results show that the harvested energy per cycle over time is improved by two orders of magnitude compared to using only a FW rectifier, and by 34 times compared to using a FW rectifier and a full-hysteresis switch in a 2-stage conditioning system. Thanks to its properties of small-size, low-weight, low-cost, and easy integration of the MEMS switch, this conditioning circuit can be easily employed to varieties of commercial energy harvesting products for large performance improvement.

## Results

**System framework**. The proposed 2-stage conditioning system (Fig. 1a) includes a TENG and a Bennet doubler as the 1st stage, a MEMS switch and a DC–DC buck converter as the 2nd stage, and a commercial regulator for stabilizing the output final voltage at 3.3 V. The high-voltage AC pulses (Fig. 1b) generated by the TENG are rectified by the Bennet at a much higher DC value than the peak-to-peak TENG output voltage. The MEMS switch is initially OFF so that the buffer capacitor ($C_{buf}$ = 4.7 nF) is charged to a high voltage (>300 V) through the Bennet (Fig. 1c). When $V_{C_{buf}}$ reaches the ON-actuation voltage ($V_{ON}$), the switch turns ON and the harvested energy in $C_{buf}$ is transferred to a high-capacitance reservoir ($C_{store}$ = 22 µF) through a 100 mH inductor. The MEMS switch is automatically actuated by the output voltage of the Bennet, while needing no additional control electronics nor power sources. The MEMS switch can be designed such a way it has not a 0 V OFF voltage, like in the previous works[22] (Fig. 1d), but it can be deactivated a few tens of volts lower than $V_{ON}$ (Fig. 1c), creating a narrow ON-OFF hysteresis and saving most of the time for recharging $C_{buf}$ to $V_{ON}$. Our system is designed to have $C_{store}$ charged to a low-voltage <18 V (Fig. 1e), so it can be followed by a commercial regulator like the LTC3588-1 to stand the output voltage at a fixed value (Fig. 1f) compatible with the requirements of portable low-power electronics.

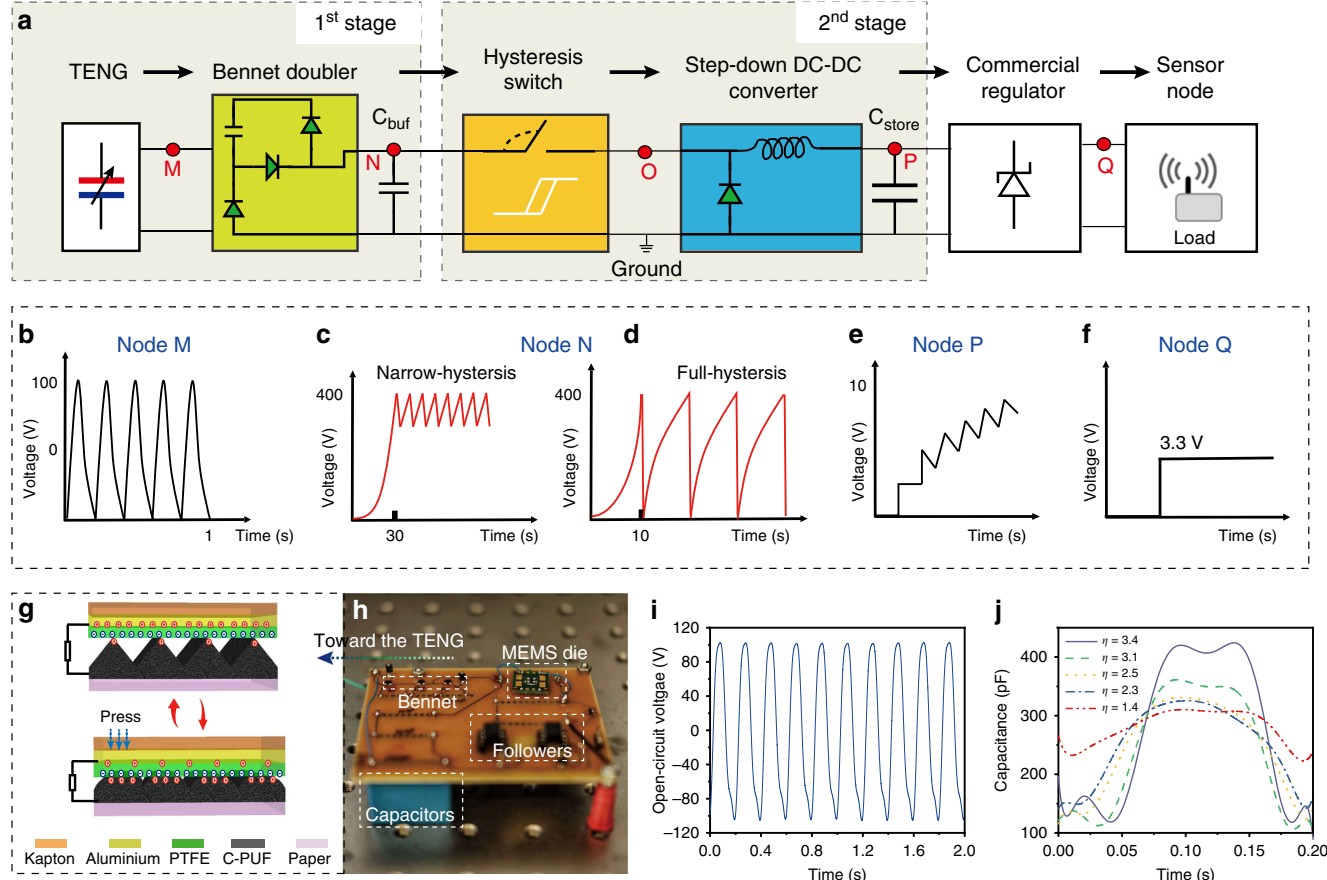

**Fig. 1 Schematic and electrical states of the conditioning circuit. a** Diagram of the conditioning system. Shapes and orders of magnitude of the voltages (**b**) generated by the triboelectric nanogenerator (TENG) used in the experiments, across the buffer capacitor with a narrow-hysteresis switch (**c**) or a full-hysteresis switch (**d**), **e** across $C_{store}$, **f** at the output of the regulator. **g** Schematic and operation principle of the TENG[44] constructed with Kapton, Polytetrafluoroethylene (PTFE), Aluminium, conductive polyurethane foam (C-PUF) and paper. **h** The printed-circuit-board of the circuit. **i** Open circuit voltage of the TENG. **j** Capacitance variations of the TENG under different tapping forces, from 2 N to 10 N with 2 N per step, corresponding to different capacitance variations ($\eta$). Source data are provided as a Source data file.

A flexible contact-separate mode TENG[44] was used whose operation principle is shown in Fig. 1g and its fabrication can be found in the method section. The main criterion for using this TENG is that its capacitance variation ratio ($\eta$) during one cycle is higher than 2, otherwise this implementation of the Bennet doubler cannot be used[45]. The TENG was activated at a frequency of 5 Hz. The implemented board and the open-circuit output voltage of the TENG are shown in Fig. 1h, i. The device shows a capacitance variations[32,47] (Fig. 1j) from $\eta = 1.4$ to $\eta = 3.4$.

**Characterization of the Bennet doubler**. The Bennet doubler has been previously applied to various electrostatic energy harvesters[28,29,44,45]. The Bennet doubler is a switched-capacitor network that amplifies the induced electrostatic charges in a TENG, implementing continuously growing rectangular charge-voltage (QV) cycles[45]. An electrical model of the Bennet doubler for TENG is shown in Fig. 2a, having a triboelectret voltage $V_{TE}$, a capacitor $C_{ref}$ (1 nF) and a buffer capacitor $C_{buf}$ (4.7 nF). The diodes are considered ideal: given the high voltages of the circuit operation, the threshold of the diodes has no impact on the circuit operation.

Figure 2b represents a simplified representation of the $i^{th}$ QV cycle of the Bennet doubler operation. Here $Q_{var}$ represents the instantaneous charge on the movable electrode of the TENG and $V_{TENG}$ the instantaneous voltage across the TENG terminals. This QV diagram starts from point A, when the TENG is already charged at its maximum capacitance $C_{max}$ and diodes $D_1$ and $D_3$

are OFF (Supplementary Fig. 1a). In the state A, all voltages are identical: $V_{TENG} = V_{C_{ref}} = V_{C_{buf}} = V_{\bar{i}}$. Here, the bar over the index $i$ indicates that the capacitance of the TENG is maximum. Thus, $C_{TENG}$ decreases at constant charge due to external mechanical forces, its voltage progressively increases until reaching $2V_{\bar{i}}$ at point B. This turns ON $D_2$ and $C_{ref}$ and $C_{buf}$ get in series with $C_{TENG}$. Because of the highest voltage across $C_{TENG}$, it gives some charges to the other two capacitors. At point D, $C_{TENG} = C_{min}$ and $D_2$ turns OFF. $C_{TENG}$ starts to decrease at a constant charge and at point E, $D_1$ and $D_3$ turn ON and the three capacitors become in parallel (diagram supported in Supplementary Fig. 1b). $V_{TENG}$ being now the lowest voltage, $C_{TENG}$ receives some charges from $C_{ref}$ and $C_{buf}$. When $C_{max}$ is reached again, the final TENG voltage at the end of the cycle $V_{\overline{i+1}}$ is slightly higher than $V_{\bar{i}}$ at the beginning of the cycle. The detailed derivations shown in Supplementary Note 1 demonstrate that the voltage expression $V_{\bar{i}}$ at the cycle $i^{th}$ is:

$$V_{\bar{i}} = \left[ 1 + \frac{C_{eq}(C_{max} - 2C_{min})}{(C_{min} + C_{eq})(C_{max} + C_{ref} + C_{buf})} \right]^i$$
$$\left( V_0 + V_{TE}\frac{C_{max} - C_{min}}{C_{max} - 2C_{min}} \right) - V_{TE}\frac{C_{max} - C_{min}}{C_{max} - 2C_{min}}$$

(1)

where $V_0$ is the initial voltage $V_{TENG}$ at $i = 0$, which in most practical cases can be considered zero. The energy converted

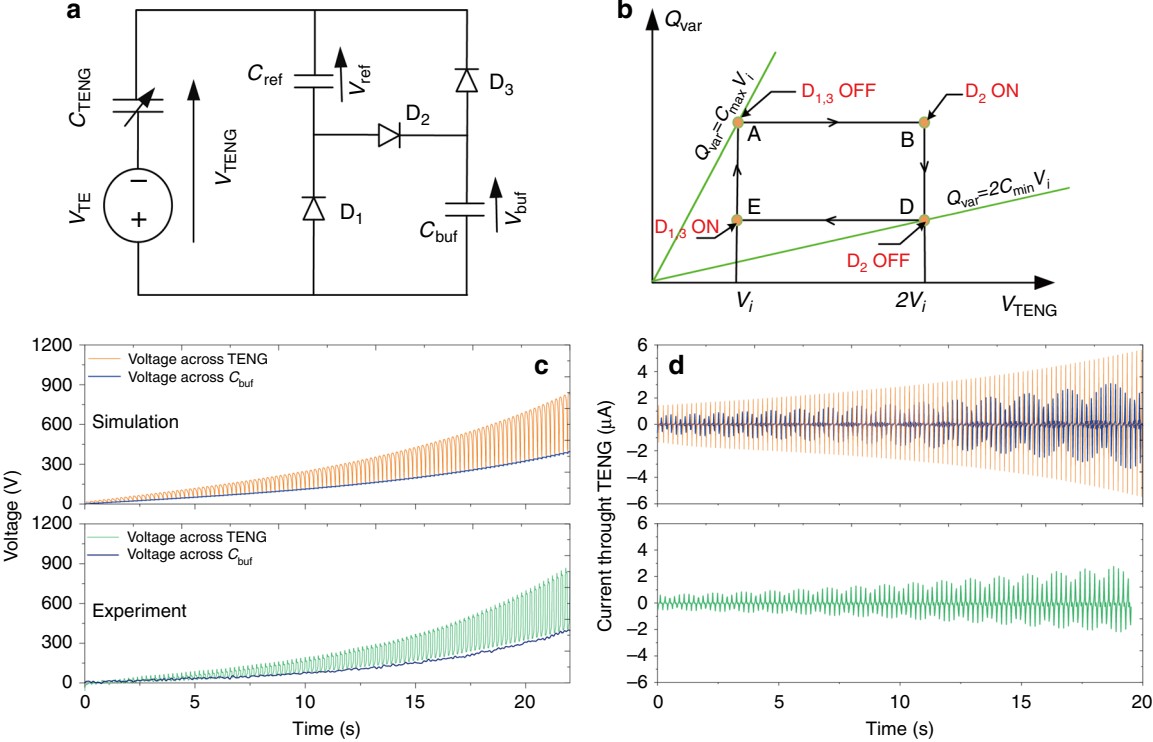

**Fig. 2 Operation principles and electrical characterizations of the Bennet doubler. a** Equivalent circuit of the TENG as well as the Bennet doubler.
**b** Theoretical QV cycle of the TENG at $i^{th}$ cycle with Bennet doubler in the steady-state. **c** Simulated and measured voltages across the TENG and $C_{buf}$.
**d** Simulated and measured currents through the TENG. In the upper figure of (**d**), the orange curve indicates a sampling rate of 10 kHz while the blue curve
indicates a low value of 50 Hz that was used in our pico-amperemeter. Source data are provided as a Source data file.

at the $i^{th}$ cycle is given by:

$$\Delta E_{\bar{i}} = \frac{1}{2}(C_{max} + C_{ref} + C_{buf})[V_{\bar{i}+1}^2 - V_{\bar{i}}^2], i \geq 1 \qquad (2)$$

It can be seen that both $V_{\bar{i}}$ and $\Delta E_{\bar{i}}$ increase exponentially with time, with a base of the exponentiation superior to 1, as far as $C_{max} > 2C_{min}$. This is a very substantial difference with the FW/HW rectifiers, where the exponential increase is always with a base of the exponentiation inferior to 1, and as a consequence, there is a limit of both voltages and converted energy[20,21,28].

The simulated and experimental results of the voltages across the TENG and $C_{buf}$, as well as the currents through the TENG, are shown in Fig. 2c, d, respectively. $V_{C_{buf}}$ equals exactly to the bottom envelope of $V_{TENG}$, and $V_{C_{buf}}$ increases exponentially, without voltage saturation. Actually, after some time, the experimental results show a saturation resulting from the Zener breakdown voltage of the diodes (not appearing in Fig. 2c). The raw data of the measured $V_{TENG}$ as well as the signal processing are given in Supplementary Fig. 2. The experimentally extracted QV cycles at different operation cycles are supported in Supplementary Fig. 3, which shows significant harvested energy per cycle (area of the QV cycle) increases as voltage goes up. This confirms that using the Bennet doubler, the harvested energy can be pumped exponentially.

Indeed, a current exponential increase of the charge transfer is observed in Fig. 2d, whereas the current peak normally keeps constant for the stable charge pumps like the FW rectifier (Supplementary Fig. 4). The experimentally measured current peaks were actually underestimated and showed some low-frequency ripples, because of the low sampling rate of the pico-amperemeter, which can be derived by comparing the simulated current curves with sampling rates of 10 kHz and 50 Hz in Fig. 2d. The harvested charges in $C_{buf}$ shown in Supplementary

Fig. 5 confirms that the exponential voltage increase with the original Bennet doubler occurs only if $C_{max} > 2C_{min}$.

**MEMS plasma switch.** To lower down the output voltage of the Bennet doubler, we have implemented a buck converter and a switch to control the energy transfer between the Bennet and the load. Until now, the bottleneck for practically operating TENGs in optimal conditions was the impossibility to operate at the same time a switch at very high-voltage, low power consumption, low current leakage and without external control. As the switch plays the role of controlling the charge transfer, plasma discharge becomes a promising solution by providing a reliable physical disconnection between electrodes, as well as high operation voltages. At a high-voltage threshold, a current flow pass through two conductive electrodes due to the electrical breakdown in a specific gas[48–51]. This principle has been used to develop macroscopic plasma sources[52–54] for TENGs. Yet, such macroscopic plasma sources were inaccurate, unstable, large-size and difficult to be controlled. Fortunately, the emerging microplasma technology that miniaturized the inter-electrode separation[55–57] made it possible to create stable plasma sources at low-vacuum pressure or even atmospheric that are otherwise only possible to be created at extreme conditions. The microplasma technique opens the door to a wide range of new exciting applications, and here we apply microplasma as a high-voltage switch to solve the pain points in the energy harvesting conditioning circuits.

With the MEMS fabrication process, a miniaturized separation between anode and cathode as low as several micrometers can be realized. According to the Paschen's law[49], the breakdown voltage increases as the electrode gap increases, except when the gap becomes too small[49], and there is a minimum breakdown voltage, ~300 V with a gap of ~5 µm[48] in air. Therefore, the ON-actuation voltage of the switch can be controlled by designing the gap

properly. However, the breakdown voltage cannot always be kept precisely constant but will be influenced by the environmental changes, i.e. humidity and temperature. Fortunately, the breakdown voltage fluctuation resulting from the humidity change is normally not larger than 10%[58,59], while the effect of the temperature on the breakdown voltage is ignorable within the industrial temperature range[60]. Concerning the device reliability and for avoiding function failure, the ON-actuation voltage should be designed with enough margin over the reverse voltage of the diodes (~570 V).

We hereby present two designs including a fixed-electrode switch with both anode and cathode fixed, and a movable-electrode switch with a movable anode and a fixed cathode. The fixed-electrode switch is wholly founded on the microplasma principle, whereas the movable-electrode switch combines the principles of micro-discharge and electrostatic pulling. The switch fabrication process is discussed in the method section.

**Fixed-electrode switch.** The fixed electrodes are designed as arrays of triangular tips (SW_F$^{Tri}$, Fig. 3a) with 80 pairs of tips, each having a side-length of 20 μm. The gap between the tips was set as 7 μm. The electric field distribution under different voltages was simulated as shown in Fig. 3b. Since the electric field threshold for the air breakdown is $3 \times 10^6$ V/m[61], the simulated breakdown voltage in our switch is estimated at 345 V as shown in Supplementary Fig. 7. Time-evolution of the voltages across the buffer capacitor ($V_{C_{buf}}$) and the storage capacitor ($V_{C_{store}}$) are shown in Fig. 3e. The switch is initially OFF and $C_{buf}$ is charged exponentially thanks to the Bennet doubler. When $V_{C_{buf}}$ reaches the threshold voltage of ~360 V, the switch turns ON and the energy stored in $C_{buf}$ is transferred to the storage capacitor ($C_{store}$) through the buck converter. A micro-discharge, corresponding to a limited charge flow, arises between electrodes. At this moment, a spark is observed (Fig. 3c, d). During the ON state, $V_{C_{buf}}$ drops

while $V_{C_{store}}$ increases, as shown in the close-up view in Fig. 3f, indicating the successful energy transfer. At ~290 V, the micro-discharge stops and the switch turns OFF, i.e. the current breaks off. A reduced hysteresis-loop (~70 V) is thus generating. Figure 3f reveals a switch ON period of 0.1 s and an OFF period of 1.3 s, denoting a switching ON/OFF frequency of 0.75 Hz.

The voltage across $C_{store}$ reaches 12 V within 55 s when the commercial regulator and the resistive load are not connected (Fig. 3e). If the regulator and load (1 MΩ) are present, the regulator wakes up for $V_{C_{store}} = 5$ V, then immediately drops to 4.5 V (Fig. 3g) which corresponds to the steady-state $V_{C_{store}}$. A maximum output current for a stable output DC voltage of 3.3 V is obtained with a 330 kΩ resistive load. Thus the practical available average power is $P_{output} = V_{dc}^2/R = 30 \mu W$, corresponding to an energy per mechanical cycle of 6 μJ as the excitation frequency of the TENG is 5 Hz.

**Movable-electrode switch.** The schematic and SEM image of the movable-electrode switch are shown in Fig. 4a. A fixed electrode as cathode is connected to Node O, i.e. the input of the buck. A movable electrode as anode is connected to Node N, i.e. the output of the Bennet ($C_{buf}$). Four suspension beams support the movement of the anode. When a high voltage is applied between the anode and the ground, the anode is pulled close to the cathode. Devices with various initial gaps $g_0$ (6 μm, 9 μm, and 12 μm) were fabricated. Due to the anode displacement, the gap between anode and cathode is no longer constant but is dynamically varying with $V_{C_{buf}}$. Thus, from the Pashen's law[49], the equation for predicting the breakdown voltage of the movable switch is given by:

$$V_{breakdown} = \frac{Bp(g_0 - x)}{\ln(Ap(g_0 - x)) - \ln\left(\ln\left(\frac{1}{\gamma} + 1\right)\right)} \quad (3)$$

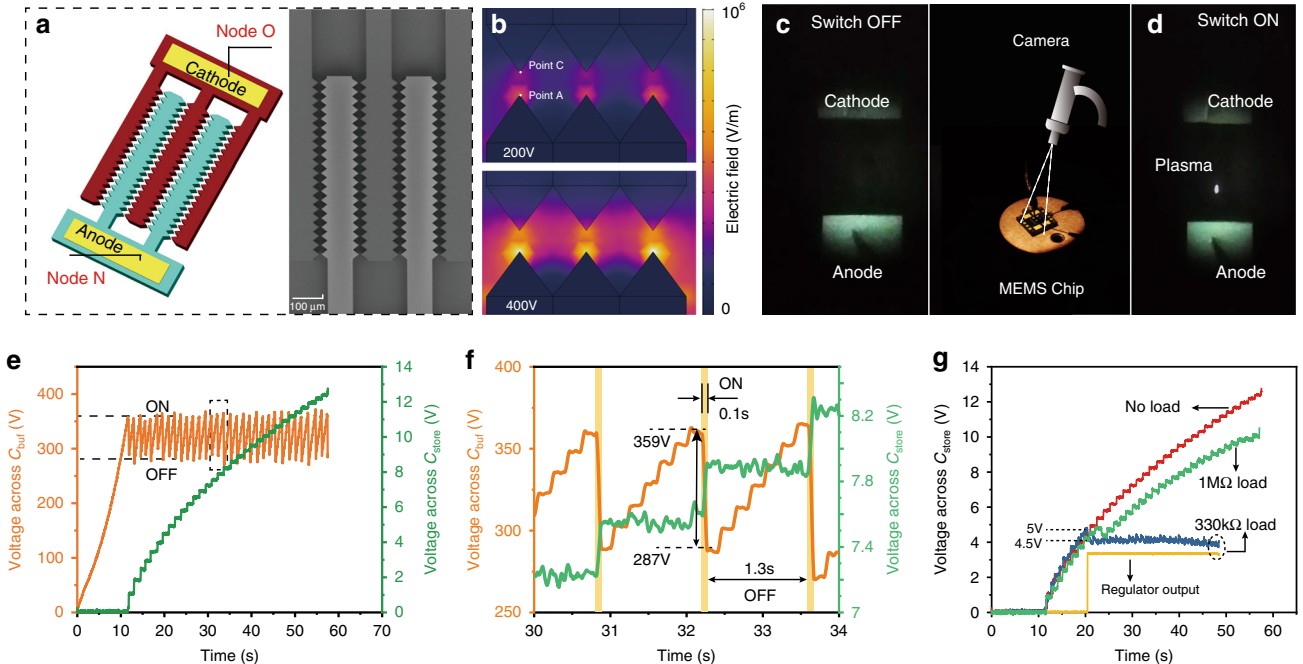

**Fig. 3 MEMS fixed plasma switch and its electrical characteristics. a** Schematic and SEM image of the MEMS fixed plasma switch. **b** The simulated electric field of three tips with voltages of 200 V and 400 V between the anode and the cathode. **c** Photo of the switch in the OFF state. **d** Photo of the switch in the ON state. The figure between **c**, **d** shows the MEMS chip and the setup to capture the plasma discharge. **e** The output voltage across $C_{buf}$ and $C_{store}$ when using the fixed switch (**a**) in the 2-stage circuit without any regulator nor load. **f** A detail of the output voltage in **e** between 30 s and 34 s. **g** Comparisons of the voltages across $C_{store}$ with different loads. Source data are provided as a Source data file.

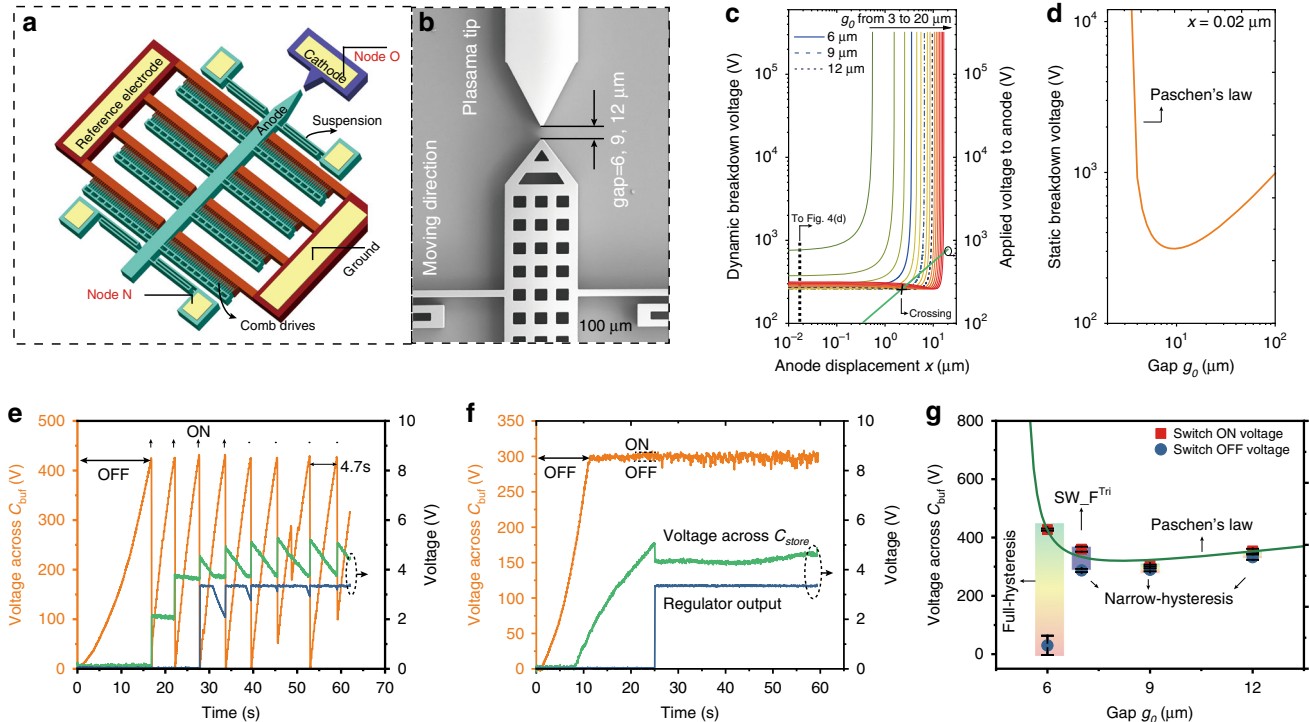

**Fig. 4 MEMS movable switch and the corresponding electrical performances. a** Schematic of the movable switch. **b** Scanning electron microscope (SEM) image of the switch. **c** Calculated dynamic breakdown voltage as functions of the anode displacement ($x$) and the relation between the voltage applied to the anode $V_{C_{buf}}$ and $x$ (blue curve). **d** Static calculated breakdown voltage versus gap $g_0$ with anode displacement of $x = 0.02\,\mu m$, which corresponds to the section of $x = 0.02\,\mu m$ in **c**. Voltage across $C_{buf}$ and $C_{store}$ and the regulator output (blue) when using a movable switch with gap 6 μm (**e**), and 9 μm (**f**) in the 2-stage system with a commercial regulator and a load of 660 kΩ. **g** The switch ON and OFF voltages versus the Paschen's law curve of silicon electrodes. Source data are provided as a Source data file.

where $g_0$ is the initial gap between anode and cathode, $x$ the displacement of the anode due to the applied voltage (see Supplementary Eq. 11) and $p$ the operating pressure. $\gamma$, $A$, and $B$ are constants related to the gas composition, the excitation-ionization energies and the saturation ionization, respectively. The calculated dynamic breakdown voltage with different initial gaps, as well as the relation between the voltage applied to the anode ($V_{C_{buf}}$) and the anode displacement are shown in Fig. 4c. A predicted breakdown voltage curve at $x \sim 0$ ($x = 0.02\,\mu m$) is shown in Fig. 4d, which corresponds to the normal Paschen's law in air. Detailed analysis of the electrostatic pulling and Paschen's law can be found in Supplementary Note 2. Seeing from Fig. 4c, there are crossing points occurring between the breakdown voltage curves of $g_0 = 9\,\mu m$, 12 μm and the related green curve of $V_{C_{buf}}$ versus $x$, while no crossing when $g_0 = 6\,\mu m$. No crossing means that the air breakdown voltage is always higher than the voltage leading to a physical contact, then the breakdown never happens. Then, the charges stored in the buffer are fully released during the contact, and a full-hysteresis loop is expected. In contrast, air breakdown happens at the crossing points (for example when $x = 2.3\,\mu m$) before the anode touches the cathode. During the breakdown, $V_{C_{buf}}$ slightly drops as the plasma current is quite low, resulting in a narrow-hysteresis loop.

In the experiments, when using a movable switch with $g_0 = 6$ μm, a full-hysteresis is obtained: the switch turns ON at a high-voltage (~425 V) and turns OFF at ~0 V, as shown in Fig. 4e. The hysteresis loop is thus ~425 V, much higher than that of the fixed switch (~70 V, Fig. 3e). The recharging time from 0 V to the ON-voltage is 4.7 s, 3.6 times longer that of the fixed switch (Fig. 3f). The switch ON-period is longer as well (0.2 s compared to 0.1 s). The ON-OFF frequency for the movable switch ($g_0 = 6\,\mu m$) is

~0.2 Hz, much lower than the fixed switch. With applying the regulator and a 660 kΩ load, the output of the regulator cannot stand at 3.3 V continuously. In contrast, when using a switch with $g_0 = 9\,\mu m$, there is a narrow ON-OFF hysteresis loop (<10 V) much lower than the fixed switch (~70 V, Fig. 3e), as shown in Fig. 4f, which results in a ON/OFF frequency of ~1.7 Hz, and the 3.3 V can now be sustained with a 660 kΩ load. The equivalent average harvested power that can be applied to the load is $P_{output} = V_{dc}^2/R = 15\,\mu W$, half to that of the fixed switch because of the lower ON voltage. The experimental results are in good match with the previous theoretical analysis that show full-/narrow-hysteresis loops with different initial gaps $g_0$ (Fig. 4g).

The multi-tip device (fixed switch) generates a higher ON current compared to the single-tip device (movable switch), while still keeping a narrow hysteresis, as shown by comparing the amplitude of the hysteresis loop in Figs. 3e and 4f. The narrower hysteresis of the movable switch ($g_0 = 9\,\mu m$) compared to the fixed switch can be explained by the fact that only one pair of triangular tips is working for the movable switch, whereas multi pairs of tips are working in the fixed switch. But it has to be noted that not all of the 80 pairs of tips are working at the same time for a specific discharge cycle. Indeed, we can see from the Supplementary Movie 1 that the flare did not appear at all of the positions of the tips, and that there is a slight difference from time to time. The other advantage the multi-tip design brought in was the improved robustness: the ON current per emitter is lower in comparison with the single-tip design[62], and if one or several tips get damaged for any reason, the other tips can still work.

**Energy transferring performance comparisons.** In order to characterize the electrical performance of the MEMS plasma

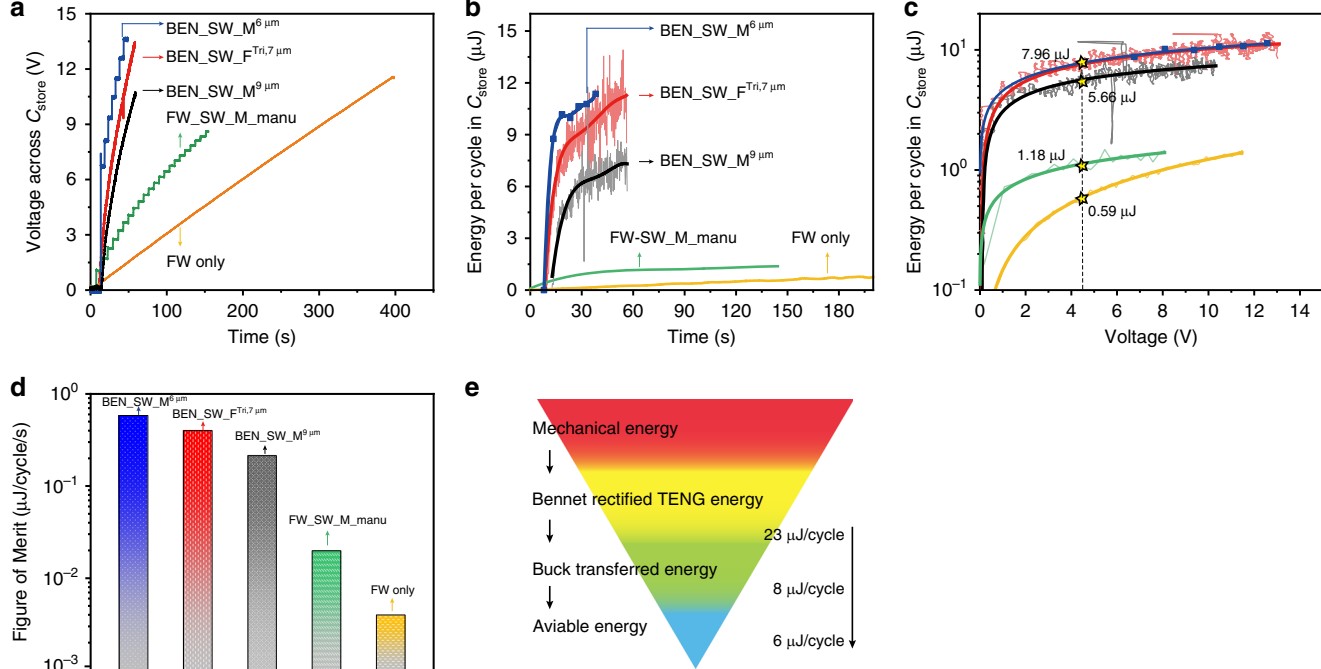

**Fig. 5 Electrical performance comparisons between different methods. a** Voltage curves of charging a capacitor of 22 µF with different conditioning methods, including the methods of: a 2-stage system using a movable switch with gap 6 µm (BEN_SW_M$^{6\,µm}$); a, using a movable switch with gap 9 µm (BEN_SW_M$^{9\,µm}$), using a fixed flat switch (BEN_SW_F$^{Tri,7\,µm}$), using manually fabricated full-hysteresis switch and FW rectifier (FW_SW_M_manu)[39]; and a classical 1-stage system using only one full-wave rectifier (FW_only). **b** Comparisons of the calculated energy per cycle versus time with different conditioning methods. **c** Comparisons of the calculated energy per cycle versus voltage under different conditioning methods. **d** Comparisons of the calculated Figure of Merit (FoM) at the voltage of 4.5 V. **e** Energy transfer flow. Source data are provided as a Source data file.

switches in the 2-stage conditioning circuit, the charging curves of $C_{store}$ under five different configurations are drawn in Fig. 5a. All the configurations are operating without commercial regulators nor loads. The harvested energy per cycle in $C_{store}$ versus time and voltage are shown in Fig. 5b, c, respectively. It shows clearly that the 2-stage circuit is much better than the conventional FW circuit, no matter what kind of switch is used. The highest charging rate and maximum harvested energy per cycle (11.3 µJ/cycle or 56.5 µW/5 Hz@13.5 V) are obtained with the 2-stage conditioning circuit and the 6 µm-movable switch due to the highest switch-ON voltage as indicated in Fig. 4e. The low ON/OFF frequency caused by the longer recharging time leads to the discontinuous output of the regulator (Fig. 4e) when using a low-impedance load. Although the high switch-ON voltage and full-hysteresis provide the fastest charging speed to $C_{store}$, it does not provide the maximum final power that can be applied to the load with a regulated voltage.

To further compare the energy harvesting efficiency together with the ability of each circuit to reach quickly the maximum energy conversion state, we defined a figure of merit (FoM) as the ratio between the energy per cycle in $C_{store}$ at the voltage of 4.5 V with the time required to reach 4.5 V from 0 V:

$$\mathrm{FoM} = \frac{\Delta E_{/cy@4.5V}}{\Delta t_{@4.5V}} \qquad (4)$$

We select 4.5 V because it is the lowest operation voltage of the regulator, as shown in Figs. 3g, and 4f. As shown in Fig. 5d, the 2-stage circuit with BEN_SW_M$^{6\,µm}$ has the highest FoM of 0.583, while the conventional FW_only circuit has the value of 0.004, indicating an improvement factor of 145. Even if considering the discontinuous final output when using BEN_SW_M$^{6\,µm}$, the FoM of the 2-stage system with BEN_SW_F$^{Tri,7\,µm}$ (0.4) is still 100 times higher than the FW_only circuit. The energy harvesting

and transfer flow of the system is shown in Fig. 5e. The efficiency of the Buck converter with the MEMS plasma switch is 35%. The energy losses can be attributed to the power consumptions of the capacitor/inductor/diode leakage and the switching dissipation.

## Discussion

We proposed a fully self-sustained MEMS high-voltage plasma switch utilizing the micro-breakdown and electrostatic pulling principles, for improving the harvesting energy efficiency in TENGs. A 2-stage conditioning circuit combined with a Bennet doubler rectifier was used for improving the charge transfer while working at very high voltages for better energy conversion. Using an unstable charge-pump like the Bennet doubler for rectifying the TENG output signal drastically reduces the constraints on the TENG efficiency, and especially on the value of its triboelectric charge density. Indeed, the Bennet doubler allows to reach any high-voltage below the dielectric breakdown, and the small-hysteresis switch maintains this high voltage at all time, whatever the triboelectric charge density of the TENG. In addition, the MEMS switch used in the buck converter doesn't need any control circuit for its actuation, which significantly decreases the total power consumption of the system. As an application of the 2-stage conditioning circuit, a 3.3 V-powered hand watch has been driven by the TENG associated with the conditioning circuits and MEMS switch (Supplementary Movie 2). The MEMS switch shows very good ON/OFF voltage stability after switching 3000 times (see Supplementary Fig. 8).

However, there is still a large space to further improve the performances of the circuit. For example, we can have more diodes in series and increase the gap to tens of or even one-hundred micrometers to approach a ~kV ON voltage. At the same time, the number of pairs of tips can be reduced to narrow the hysteresis while keeping a proper redundancy of the tips to

ensure the robustness and sustainability of the switch. The MEMS device used in this paper is exposed in air, which can be easily polluted by the dusts and become invalid if there is water condensation between the electrodes, causing conduction at a much lower voltage, without any dielectric breakdown. Therefore, for a practical application, a hermetic package is preferable. This can even be obtained at the wafer scale (batch process) by performing an anodic bonding with a glass wafer[63,64].

The silicon devices used in this paper can contain the current limits in most of the energy harvesting applications, because the average ON-current is in orders of µA, far below the safe current of tens of mA[50]. The MEMS silicon fabrication process holds the advantages of simple process, low-cost, and batch fabrication. However, the tradeoff is the relatively poor electrical conductivities of the material and the thin oxide layer that grows on the surface which limits the current. For our application in energy harvesting, it may decrease the energy transfer efficiency. Microdischarges could also vaporize or sputter the electrode material after a long-term running at high current[48], which increases the pressure in the gap spacing if the device is packaged and changes the gap itself, thus decreases the breakdown voltage. Employing tough metal materials would help to improve the emission efficiency and stability, for example, tungsten coating on silicon-based gated emitters[65,66]. However, we have to notice that such kind of tungsten coating or metal deposition will decrease the turn-on/breakdown voltage[48], on the contrary with our purpose of increasing the operation voltage as high as possible. Therefore, the system can benefit from the good conductivity and low leakage of the metal materials, only on the basis of proper design for not decreasing the breakdown voltage. To definitely ensure that no reliability issues would occur after a long period of actuation, current regulators might be added to protect the silicon tips of the electrodes. This can be obtained by integrating a negative current feedback, i.e. having some current limiters in series with the silicon tips, as proposed in[67], where thin and long silicon tips implemented ungated MOS transistors providing a current limitation.

In conclusion, the proposed MEMS switch shows promises not only for the conditioning circuits of triboelectric nanogenerators but also for a wide range of energy harvesters. For instances, the specifically designed MEMS switches can be applied to control the synchronized switch harvesting on inductor circuits[9] which are used to improve the harvesting efficiency of electret and piezoelectric energy harvesters. The employment of the MEMS switch in the conditioning circuits can significantly push forward the practical and commercial applications of the energy harvesters by largely improving the systematic performance.

## Methods

**Fabrication of the TENG**. The TENG was fabricated based on a macrostructured conductive PU foam (C-PUF) doped with conductive carbon black powder, and a polytetrafluoroethylene (PTFE) film. The C-PUF was shaped into macro scale triangle prisms, while its porous surface has micro/nano 3D structures[44]. At the same time, the C-PUF plays the roles of spring, spacer, friction layer, and electrode. First, an aluminium electrode with a dimension of 60 mm × 60 mm × 100 µm was pasted to an insulating Kapton substrate. Then a PTFE film of 50 µm was sticked to the aluminium electrode, as the negative triboelectret layer. The second electrode of the device was fabricated using the C-PUF with initial thickness of 10 mm, manually shaped with 5 triangle prisms. The C-PUF was sticked to a paper substrate. The fabricated device was placed under a vibration shaker (MODAL SHOP K2007E01), where the vibration frequency and amplitude/force can be controlled with a signal generator (Tektronix AFG3102).

**Fabrication of the switch**. The switch was fabricated in a silicon-on-insulator wafer with device layer of 40 µm in thickness. A dicing-free process[68] was applied to improve the yield rate of the movable switches. The fabrication process mainly includes the following steps: (a) photolithography for front-side DRIE and pads opening; (b) front DRIE for the pads opening; (c) remove photoresist and aluminum sputtering (1 µm); (d) remove the backside oxidation using buffered oxide

etching; (e) backside aluminum sputtering (0.5 µm); (f) front DRIE etching until the box layer; (g) backside DRIE etching until the box layer; (h) vapour HF release, as supported in Supplementary Fig. 9.

**Measurement of the high voltage**. The voltage across the buffer capacitor is out of the limitation of the follower amplifier, thus we developed a capacitor divider, which has two capacitors (100 nF and 5 nF) in series, as supported in Supplementary Fig. 10. $C_{res}$ is a series combination of $C_{res1} = 5$ nF and $C_{res2} = 100$ nF thus the voltage applied on $C_{red}$ is calculated by: $V_{C_{res}} = V_{osc}(1 + (C_{res2}/C_{res1}))$, where $V_{OSC}$ is the voltage measured with the oscilloscope. By measuring the voltage across the 100 nF capacitor, the voltage across the small capacitor can be calculated by multiplying it by a factor of 21. Each diode in the circuits reports two diodes with 285 V inverse voltage in series. It is used to overcome the voltage limitation of the follower since two diodes can enlarge the reverse voltage from 285 V to 570 V.

## Data availability

The source data underlying Fig. 1h–i; 2c, d; 3e–g; 4c–g; and 5a–d; and Supplementary Figs. 2–4; 5a, b; 7 and 8 are provided as a Source data file. Extra data are available from the corresponding author upon reasonable request.

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

## Author contributions

H.Z. and P.B. conceived the project. F.M. developed the MEMS process and fabricated the device. H.Z., D.G., and P.B. designed the experiments, performed the electrical performance measurement and wrote the paper. X.X., Y.Z., and T.B. provided some suggestions on the electrical measurement. All authors discussed the results and contributed to the improvement of the paper.

## Competing interests

The authors declare no competing interests.
