## [Peer Review File · Nature Communications]

Reviewers' comments:

Reviewer #1 (Remarks to the Author):

The manuscript reported an efficient energy management system for TENGs using a MEMS plasma switch. The energy harvesting efficiency is improved two orders of magnitude compared to a full-wave rectifying circuit. TENGs are new energy harvesting devices that can produce high voltage but low current, and the energy management has been a challenge. This work provide a feasible solution and is of interests to researchers in the field. I recommend the publication of this manuscript. The manuscript has some minor errors and author might want to fix them. For example, the caption of fig 3. "The figure between (g) and (h)" should be "The figure between (d) and (e)".

Reviewer #2 (Remarks to the Author):

This an interesting paper, showcasing a promising approach to implement useful energy harvesters harnessing the triboelectric effect. The increase in harvesting efficiency of two orders of magnitude is significant. Please take care of the following issues to make your paper stronger and acceptable for publication:

1-There seems to be a reference missing, the reference should fall between Ref. [35] and [36]. You quote it on page 2 and 5. Please look into and either add the reference or remove it (so the word processor does not show an error).

2-The authors should comment on the suitability of their approach (microplasma switch) to environmental changes, e.g. humidity. The switch seems to need some kind of specialized packaging, e.g. hermetic, inert atmosphere. What would be the specifications of this controlled environment? please comment.

3-Page 7, Fig. 2 (c) and (d). Please comment on why your modeling underestimates the voltage and overestimates the current (the opposite effects are good, they try to compensate each other when calculating power). Also, it is good that modeling and experiment agree on the lower bound (what you call "bottom envelope"), but there are features in the experimental data that are not captured by your model, e.g. the current ripples that seem to have a slower timescale than the natural oscillations Fig. 2(d). Where does this come from? Is this an artifact in the experiment or is inherent to your approach? please explain.

4-page 8. You are missing a paragraph on microplasmas to provide context. In nutshell, by miniaturizing the inter-electrode separation, plasma sources can operate stably at less vacuum, even at atmospheric pressure, which opens very exciting opportunities, e.g. creating excited species that are otherwise only possible to create at extreme conditions. You should also some key references on micro plasmas. I suggest to include these:

-K H Becker et al 2006 "Microplasmas and Applications", J. Phys. D: Appl. Phys. 39 R55

and key recent work (focused on agile manufacturing, but still relevant to the microplasma "big picture")

-Y S Kornbluth et al 2018 "Microsputterer with integrated ion-drag focusing for additive manufacturing of thin, narrow conductive lines" J. Phys. D: Appl. Phys. 51 165603

-S. Ghosh et al. 2014 "Fabrication of Electrically Conductive Metal Patterns at the Surface of Polymer Films by Microplasma-Based Direct Writing" ACS Appl. Mater. Interfaces, 6, 5, 3099-3104

5-Page 8. Spring softening (the spring constant is reduced) due to gap reduction is a well known phenomenon in MEMS mechanical switches. How spring softening affects (or improves?) your approach? please explain.

6-Somewhere in the text, the authors should clearly state that solid-state switches are not good for this application because they are inherently leaky. You need a mechanical switch that physically disconnects the circuit.

7-Please comment on reliability issues. Your devices are made of Si. Are you concerned about lifetime? would it help to make the devices in other materials, e.g. tungsten? please help us understand the trade-offs, e.g. energy function, fatigue, stability of physical properties (e.g. single crystal vs, multi-crystal). This is a significant issue for your technology to be adopted as mainstream.

8-Based on your data, the choice of $g_0 = 9 \text{ } \mu\text{m}$ seems to be a fortunate coincidence; what would it take to optimize the design? In other words, maybe there is a better value for g_0 , with even better performance. What would it take to find it?

9-Page 11, you mention that 80 pairs of tips are working in the fixed-fixed switch. I respectfully disagree. The microplasma is a non-linear phenomenon, it is very sensitive to the tip radii (tip electric fields depend on their tip radius), my guess is that only a few of them are working during the discharge due to the tip radii spread (it is unavoidable when you make arrays of anything, and when one makes arrays of very small features, the spread tends to have long tails).

10-Talking about the multi-tip switch, why do you have so many tips? are you concerned about lifetime (so as soon as the sharper tips get damaged, the duller tips start working)? please comment

11- Supplementary video 2. Maybe is my browser, but your video is upside down! can you please check you uploaded the video in the right orientation?

12-Thinking some more about your multi-tip device, the authors should point out that you might need current regulators in each emitter to uniformize the current in the device, to protect the tips from burning/damaging, to efficiently use the array of tips. In a nutshell, you can take care of the tip radii spread if you integrate negative feedback, i.e., having some ballast in series with each tip, so there is a maximum current per emitter, so the sharper tips can work at the same time the duller tips work. An ideal ballast has high impedance and high saturation current, e.g. an ungated field effect transistor (FET) operating in saturation, which is not hard to achieve given the voltage

involved (you can get the FETs to saturate with volts). Given that you are making your structures in silicon, you can easily make long and narrow fingers ending in tips, which would make the fingers act as ungated FETs monolithically integrated to the sharp tips. Researchers have reported a similar idea for field emission of electrons:

-L. F. Velasquez-Garcia et al. 2011 "Uniform High-Current Cathodes Using Massive Arrays of Si Field Emitters Individually Controlled by Vertical Si Ungated FETs—Part 1: Device Design and Simulation" IEEE Transactions in Electron Devices, vol. 58, No. 6, pp. 1775 - 1782.

please add such discussion. Again, this is a significant issue for your approach to be adopted mainstream.

Rebuttal letter

High-efficiency Conditioning System for High-Voltage Management in Triboelectric/Electrostatic Kinetic Energy Harvesters Employing a MEMS Plasma Switch

Hemin Zhang^{1,2}, Frédéric Marty¹, Xin Xia³, Yunlong Zi³, Tarik Bourouina¹, Dimitri Galayko^{4*}, and Philippe Basset^{1*}

¹ ESYCOM, Univ Gustave Eiffel, CNRS, CNAM, ESIEE Paris, F-77454 Marne-la-Vallée, France.

² Department of Engineering, The Nanoscience Centre, University of Cambridge, Cambridge CB3 0FF, United Kingdom.

³ The Chinese University of Hong Kong, Shatin, N.T., Hong Kong SAR, China

⁴ Sorbonne Université, LIP6, France.

*Correspondence to: philippe.basset@esiee.fr, dimitri.galayko@sorbonne-universite.fr

We appreciate the positive comments and constructive suggestions made by the reviewers. We have revised our manuscript according to these comments. In order to make the revision clear and easy to read, we marked our change in the manuscript using different colors, and we also listed them below one by one.

Color instructions:

Blue in this rebuttal letter: response to the reviewer

Red in this rebuttal letter, the main document and the supplementary material: added new materials according to the comments

Blue in the main document: added new materials by the authors

Green in the main document: data source illustration

Reviewer #1

The manuscript reported an efficient energy management system for TENGs using a MEMS plasma switch. The energy harvesting efficiency is improved two orders of magnitude compared to a full-wave rectifying circuit. TENGs are new energy harvesting devices that can produce high voltage but low current, and the energy management has been a challenge. This work provides a feasible solution and is of interests to researchers in the field. I recommend the publication of this manuscript. The manuscript has some minor errors and author might want to fix them. For example, the caption of fig 3. "The figure between (g) and (h)" should be "The figure between (d) and (e)".

Response: We appreciate the positive comments from the reviewer. The manuscript has been thoroughly checked and these kinds of minor errors have been corrected.

Reviewer #2

This is an interesting paper, showcasing a promising approach to implement useful energy harvesters harnessing the triboelectric effect. The increase in harvesting efficiency of two orders of magnitude is significant. Please take care of the following issues to make your paper stronger and acceptable for publication:

Response: We greatly appreciate the positive comments, professional questions and helpful suggestions from the reviewer.

Q1- There seems to be a reference missing, the reference should fall between Ref. [35] and [36]. You quote it on page 2 and 5. Please look into and either add the reference or remove it (so the word processor does not show an error).

Response: The references were corrected and thoroughly checked in the revision. We also improve the introduction for a better understanding (changes are marked in blue).

Q2- The authors should comment on the suitability of their approach (microplasma switch) to environmental changes, e.g. humidity. The switch seems to need some kind of specialized packaging, e.g. hermetic, inert atmosphere. What would be the specifications of this controlled environment? please comment.

Response: The environmental changes like the humidity surely have impacts on the approach, by influencing the breakdown voltage of the plasma switch. Learning from the previous research [67], we know that water vapor has a higher breakdown strength than air, so a mixture of the water vapor and air (i.e. higher humidity) results in a higher breakdown voltage. Fortunately, previous researches have confirmed that this kind of breakdown voltage fluctuation was normally not higher than 10% when the relative humidity is lower than 70% [68, 69]. For our approach, a little breakdown voltage (ON actuation voltage) increase is good for the performance improvement as we are always pursuing a high operation voltage. The important point that needs to be paid attention to is that the switch breakdown voltage should not be higher than the reverse voltage of the diodes. In our design, the switch breakdown voltage was between ~300V and ~420V, which was far away from the diode limit of ~570V. Thus we believe that reasonable humidity change will not have a big impact on the suitability of the approach.

Compared to humidity, temperature has much less impact on the breakdown voltage if the temperature of operation is within the industrial temperature [70][71].

Contrariwise, if there is water condensation between the two electrodes of the switch, conduction could occur at a much lower voltage, without any dielectric breakdown. In that case the system won't work correctly. Then, for a practical application, a hermetic package is preferable, also to protect the tips from dust pollution. This can even be obtained at the wafer scale (batch process) by performing an anodic bonding with a glass wafer. Combined with getter material, it also can also provide a good long-term vacuum [74][75]. We added related comments to the manuscript.

In the "The MEMS plasma switch" section (p. 8):

"Therefore, the ON-actuation voltage of the switch can be controlled by designing the gap properly. However, the breakdown voltage cannot always be kept precisely constant but will be influenced by the environmental changes, i.e. humidity and temperature. Fortunately, the breakdown voltage fluctuation resulting from the humidity change is normally not larger than 10% [67][68][69], while the effect of the temperature on the breakdown voltage is ignorable within the industrial

temperature range [70][71]. Concerning the device reliability and for avoiding function failure, the ON-actuation voltage should be designed with enough margin over the reverse voltage of the diodes (~570V).”

In the “Discussion” section (p.14):

“The MEMS device used in this paper is exposed in air. It can be easily polluted by the dusts and become invalid if there is water condensation between the two electrodes of the switch, causing conduction between electrodes at a much lower voltage, without any dielectric breakdown. Therefore, for a practical application, a hermetic package is preferable. This can even be obtained at the wafer scale (batch process) by performing an anodic bonding with a glass wafer. Combined with getter material, it can also provide a good long-term vacuum [74][75].”

- [67]. M. Radmilović-Radjenović, B. Radjenović, Ž. Nikitović, Š. Matejčik, M. Klas, The humidity effect on the breakdown voltage characteristics and the transport parameters of air. Nuclear Instruments and Methods in Physics Research Section B: Beam Interactions with Materials and Atoms. 279, 103-5 (2012).
- [68]. E. Ikuffe, Influence of humidity on the breakdown voltage of sphere-gaps and uniform-field gaps. Proceedings of the IEE-Part A: Power Engineering. 108, 295-301 (1961).
- [69]. K. R. Allen, K. Phillips, Effect of humidity on the spark breakdown voltage. Nature 183, 174-5 (1959).
- [70]. H. Fujita, T. Kouno, Y. Noguchi, S. Ueguri, Breakdown voltages of gaseous N₂ and air from normal to cryogenic temperatures. Cryogenics. 18, 195-200 (1978).
- [71]. W. S. Zaengl, S. Yimvuthikul, G. Friedrich, The temperature dependence of homogeneous field breakdown in synthetic air. IEEE transactions on Electrical insulation. 26, 380-90 (1991).
- [74]. O. Gigan, H. Chen, O. Robert, S. Renard, F. Marty, Fabrication and characterization of resonant SOI micromechanical silicon sensors based on DRIE micromachining, freestanding release process and silicon direct bonding. International Society for Optics and Photonics in Nano-and Microtechnology: Materials, Processes, Packaging, and Systems, 4936, 194-204 (2002).
- [75]. B. Lee, S. Seok, K. Chun, A study on wafer level vacuum packaging for MEMS devices. Journal of micromechanics and microengineering. 13, 663, (2003).

Q3- Page 7, Fig. 2 (c) and (d). Please comment on why your modeling underestimates the voltage and overestimates the current (the opposite effects are good, they try to compensate each other when calculating power). Also, it is good that modeling and experiment agree on the lower bound (what you call "bottom envelope"), but there are features in the experimental data that are not captured by your model, e.g. the current ripples that seem to have a slower timescale than the natural oscillations Fig. 2(d). Where does this come from? Is this an artifact in the experiment or is inherent to your approach? please explain.

Response: Thank you for pointing out this disparity between simulations and experiments. Indeed, additional information is needed for a good understanding of these apparent mismatches: actually lower voltage bound, upper voltage bound and current ripples can be correctly caught by simulations.

a) Concerning the higher amplitude of the TENG’s voltage in Fig 2c:

If we do a zoom in of a few cycles of the experimental curve, we observe a peak at each front edge followed by a damped oscillation at 50Hz:

Figure R1

Raw data of the measured voltage of the TENG output after a $1\text{G}\Omega/70\text{G}\Omega$ voltage divider showing a peaks at the front edges and 50Hz ripples.

The voltage peak is directly induced by the setup we used for measuring the high-voltage, i.e. a $1/71$ resistive divider made of 2 resistors of $70\text{G}\Omega$ and $1\text{G}\Omega$. Indeed, it comes from the parasitic capacitance of the $70\text{G}\Omega$ resistor, which shows a low impedance compared to the resistance for this voltage having such a high slope. We can simulate this peak by adding a 100fF capacitor in parallel to the $70\text{G}\Omega$ resistor, as shown below:

Figure R2

(a) Simulation of the voltage across the TENG
(b) Simulation of the TENG output voltage after the $1\text{G}/70\text{G}$ resistive divider including a parasitic capacitor of 100fF across the 70G resistor.

As the peak is quite sharp, it doesn't contribute significantly on the output average current nor the harvested energy. For the new experimental curve in Fig. 2c, we have added a low-pass filter at 40Hz for attenuating the electromagnetic noise, followed by a 50th percentile filter of 65 points for smoothing the peak induced by the parasitic capacitance. The raw data and the post-process operations are presented in the Supplementary Materials Fig. S2.

Figure S2. The experimentally measured voltage across the TENG. (a) Raw measured data. **(b)** Measured data after applying a 40Hz low-pass filter. **(c)** Low-pass filtered data with a 50%-percentile smoothing of 65 points.

In the revision, we replaced the experimental data in Fig. 2c with the smoothed data and added the corresponding instructions in the main manuscript in page 6 marked in red. “The raw data of the measured V_{TENG} as well as the signal processing are given in Supplementary Materials Fig. S2.”

b) Concerning the ripples in the TENG’s current in Fig. 2d

The low frequency ripple of the measured current comes from the low sampling rate (50Hz) of the used picoamperemeter, which is far below the Nyquist rate associated to this signal. So, the full signal is not captured and a low frequency ripple is introduced.

We can simulate this ripple by resampling at 50Hz the simulated current through the TENG, whose sampling rate is 10kHz. In the revision, we added both simulated curves with different sampling rates in Fig. 2d, and added the corresponding descriptions in the main manuscript in page 7 marked in red.

“The experimentally measured current peaks were actually underestimated and showed some low-frequency ripples, because of the low sampling rate of the picoamperemeter, which can be derived by comparing the simulated current curves with sampling rates of 10kHz and 50Hz in Fig. 2d.”

Figure 2 | Operation principles and electrical characterizations of the Bennet doubler. (a) Equivalent circuit of the TENG as well as the Bennet doubler. **(b)** Theoretically QV cycle of the TENG at i^{th} cycle with Bennet doubler in steady-state. **(c)** Simulated and measured voltages across the TENG and C_{buf} . **(d)** Simulated and measured currents through the TENG. C_{ref} and C_{buf} are set as 1nF and 4.7nF respectively. In the upper figure, the orange curve indicates a sampling rate of 10kHz while the blue curve indicates a low value of 50Hz that was used in our picoamperemeter.

Q4- page 8. You are missing a paragraph on microplasmas to provide context. In nutshell, by miniaturizing the inter-electrode separation, plasma sources can operate stably at less vacuum, even at atmospheric pressure, which opens very exciting opportunities, e.g. creating excited species that are otherwise only possible to create at extreme conditions. You should also some key references on micro plasmas. I suggest to include these:

-K H Becker et al 2006 "Microplasmas and Applications", J. Phys. D: Appl. Phys. 39 R55 and key recent work (focused on agile manufacturing, but still relevant to the microplasma "big picture")

-Y S Kornbluth et al 2018 "Microsputterer with integrated ion-drag focusing for additive manufacturing of thin, narrow conductive lines" J. Phys. D: Appl. Phys. 51 165603

-S. Ghosh et al. 2014 "Fabrication of Electrically Conductive Metal Patterns at the Surface of Polymer Films by Microplasma-Based Direct Writing" ACS Appl. Mater. Interfaces, 6, 5, 3099-3104

Response: We added sentences to describe the microplasma technology and claimed our energy harvesting circuit using the micro-plasma switch to be one of the new exciting applications of the micro-plasma technology. The suggested papers are cited with index numbers of [64][65][66]. Please refer to page 8, paragraph 1, marked in red.

“As the switch in the conditioning circuit plays the role of controlling charge transfer, plasma discharge becomes a promising solution by providing reliable physical disconnection between electrodes, as well as high operation voltages. At a high-voltage threshold, a current flow will pass through two conductive electrodes due to the electrical breakdown in a specific gas [57][58][59][60] (air or some rare gases). This principle has been previously used to develop manually-fabricated macroscopic plasma sources [61][62][63] for TENGs. However, in such a macroscopic scale, the plasma sources were inaccurate, unstable, large-size, and difficult to be controlled. Fortunately, the emerging of the microplasma technology that has miniaturized the inter-electrode separation [64][65][66] made it possible to create stable plasma sources at low-vacuum pressure or even atmospheric that are otherwise only possible to be created at extreme conditions. Such kind of microplasma technique opens the door to a wide range of new exciting applications. Here we first apply microplasma as a high-voltage switch to solve a bottleneck in the energy harvesting conditioning circuits.”

[64]. K. H. Becker, K. H. Schoenbach, J. G. Eden, Microplasmas and applications. Journal of Physics D: Applied Physics 39, 55-70 (2006).

[65]. Y. S. Kornbluth, R. H. Mathews, L. Parameswaran, L. M. Racz, L. F. Velásquez-García, Microsputterer with integrated ion-drag focusing for additive manufacturing of thin, narrow conductive lines. Journal of Physics D: Applied Physics. 51, 165603 (2018).

[66]. S. Ghosh, R. Yang, M. Kaumeyer, C. A. Zorman, S. J. Rowan, P. X. Feng, R. M. Sankaran, Fabrication of electrically conductive metal patterns at the surface of polymer films by microplasma-based direct writing. ACS applied materials & interfaces 6, 3099-104 (2014).

The descriptions of the MEMS plasma switch in the following paragraph “Fixed-electrode switch” were also revised as a logical consequence, marked in blue in the revised manuscript.

Q5- Page 8. Spring softening (the spring constant is reduced) due to gap reduction is a well known phenomenon in MEMS mechanical switches. How spring softening affects (or improves?) your approach? please explain.

Response: Thank you for this question. Regarding the softening effect, we believe that the electrostatic softening effect for our device is not very obvious as comb-drives are used and the movement direction is longitudinal. The device works more like an electrostatic actuator rather than a resonator. The simulated resonant frequency of the device is $\sim 13\text{kHz}$ while the actual movement frequency of the device actuated by the electrostatic force is lower than 2Hz . The frequency-amplitude effect caused by the spring softening can thus be ignored.

However, it is still valuable to explain clearly the dynamics and working principles of the movable switch, whose breakdown voltage is always varying because of the movement of the anode, i.e. dynamic changes of the gap between anode and cathode. In this revision, we added more simulations and theoretical discussions to explain why we have a full-hysteresis loop when $g_0 = 6\mu\text{m}$ and a narrow hysteresis loop when $g_0 = 9\mu\text{m}$ and $12\mu\text{m}$. Please refer to the following description that is added in page 10 and 11, marked in red.

Figure 4 | Design of the MEMS movable switch and the corresponding electrical performances. (a) Schematic of the movable switch. **(b)** SEM images of the switch. **(c)** Calculated dynamic breakdown voltage as functions of the anode displacement (x) and the relation between the voltage applied to the anode $V_{C_{buf}}$ and x (blue curve). **(d)** Static calculated breakdown voltage versus gap g_0 with anode displacement of $x=0.02\mu\text{m}$. This curve corresponds to the section of $x=0.02\mu\text{m}$ in (c). Voltage across C_{buf} and C_{store} and the regulator output when using a movable switch with gap $6\mu\text{m}$ **(e)**, and $9\mu\text{m}$ **(f)** in the 2-stage system with a commercial regulator and a load of $660\text{k}\Omega$. **(g)** The switch ON and OFF voltages versus the Paschen's law curve of silicon electrodes

“Due to the movement of the anode, the gap between anode and cathode is no longer constant but is dynamically varying with the output voltage of the Bennet’s doubler. Thus, from the equation of the Paschen’s law [60], equation for predicting the breakdown voltage of the movable switch:

$$V_{breakdown} = \frac{Bp(g_0 - x)}{\ln(Ap(g_0 - x)) - \ln(\ln(\frac{1}{\gamma_i} + 1))} \quad (3)$$

where g_0 is the designed initial gap between anode and cathode, x the movement of the anode due to the applied voltage (see Supplementary Materials eq. S11), p the operating pressure, γ_i , A and B are constants related to the gas composition, the excitation-ionization energies and the saturation ionization respectively. The calculated dynamic breakdown voltage with different initial gaps between anode and cathode, as well as the relation between the voltage applied to the anode i.e. $V_{C_{buf}}$ (eq. S11) and the anode displacement are shown in Fig. 4c. A predicted breakdown voltage curve at $x=0$ ($x=0.02\mu\text{m}$) is shown in Fig. 4d, which corresponds to the normal Paschen's law in air. Detailed theoretical analysis of the electrostatic pulling and Paschen's law can be found in section III of Supplementary Materials. Seeing from Fig. 4c, there are crossing points between the breakdown voltage curves of $g_0 = 9\mu\text{m}$, $12\mu\text{m}$ and the related green curve of $V_{C_{buf}}$ versus x , while no crossing when $g_0 = 6\mu\text{m}$ occurs. If there is no crossing, it means that the air breakdown voltage is always higher than the voltage leading to a physical contact, then the breakdown will never happen. The charges stored in the buffer will be fully released ($V_{C_{buf}}$ drops to 0V) during the contact, thus a full-hysteresis loop is expected. In contrast, air breakdown happens at the crossing points (for example when $x = 2.3\mu\text{m}$) before the anode touches the cathode. During the breakdown, $V_{C_{buf}}$ will slightly drop as the plasma current is quite low, resulting in a narrow-hysteresis loop."

Q6- Somewhere in the text, the authors should clearly state that solid-state switches are not good for this application because they are inherently leaky. You need a mechanical switch that physically disconnects the circuit.

Response: We thank the reviewer for this suggestion. The corresponding comments are added in the introduction, page 3, paragraph 2, marked in red.

"Besides, additional energy dissipation will be brought in because this kind of solid-state electronic switch is inherently leaky. Superior electrical insulation between the two stages is mandatory, therefore an acceptable switch must have good physical disconnection properties when the switch is OFF."

Correspondingly, when stating the advantages of our micro-plasma switch, we have added the property of "no ohmic contact" as marked in red in the following paragraph in page 4.

"Compared to the electronic switches, the proposed micro-plasma switch has the advantages of no electronic control, no ohmic contact and no need to be supplied with some external energy. The proposed switch is a fully "stand-alone" device and does not require direct integration with the TENG."

Q7- Please comment on reliability issues. Your devices are made of Si. Are you concerned about lifetime? would it help to make the devices in other materials, e.g. tungsten? please help us understand the trade-offs, e.g. energy function, fatigue, stability of physical properties (e.g. single crystal vs, multi-crystal). This is a significant issue for your technology to be adopted as mainstream.

Response: The dominant point we concern in this paper is to verify the effectiveness of the MEMS plasma switch used in energy harvesters to improve the energy harvesting efficiency. It is well-known that silicon-on-insulator (SOI) fabrication has the advantages of simple process, low-cost, and batch fabrication. Thus we choose this simplest and fastest fabrication solution that we are good at, i.e. etching tips with single-crystal silicon, to demonstrate a basic energy-harvesting system.

Regarding the reliability of the silicon switch, our major concern when we did the design was avoiding having a current so high it could damage the silicon tips. Micro-discharges could vaporize or sputter the electrode material after a long-term running at high current [57], which increases the pressure in the gap spacing if device is packaged and changes the gap itself, thus decreases the breakdown voltage. According to [59], a continuous current as high as tens of mA will cause physical damage and function failure for a silicon microplasma device. Fortunately, in our experiments, we calculated the average emission current in orders of μA . For example, in Fig. 3f, the average current can be calculated as $i=Q/t=\Delta V \cdot C_{\text{buf}}/\Delta t=70\text{V} \cdot 4.7\text{nF}/0.1\text{s}=3.3\mu\text{A}$. In addition, in some devices, the charge transfer is distributed on multi pairs of tips (possibly up to 80 pairs). If we consider only 10 pairs of tips working simultaneously, the current through each pair of tips is about $0.3\mu\text{A}$. Therefore, in terms of the current limit, our silicon MEMS switch can fully fulfill the requirements of our application.

However, the shortages of using silicon are also obvious, including the poor electrical conductivities of the material and the thin oxide layer that grows on the surface that can limit the current. For our application in the energy harvesting field, it may decrease the energy transfer efficiency because of the energy consumption caused by the poor electrical conductivity if a highly doped substrate is not used.

Employing tough metal materials will help to improve the emission efficiency and stability. For example, tungsten coating on silicon-based gated emitters have been used [76][77]. However, we have to notice that such kind of tungsten coating or metal deposition will decrease the turn-on/breakdown voltage [57], which is on the contrary with our purpose of having the operation voltage as high as possible. Therefore, the system can benefit from the good conductivity and low leakage of the metal materials, only on the basis of not decreasing to much the breakdown voltage.

The corresponding comments were added to the discussion part (p.15, marked in red):

“The single-crystal silicon devices used in this paper can contain the current limits in most of the energy harvesting applications, because the calculated average ON-current is in orders of μA far below the safe current of tens of mA [59]. The MEMS silicon fabrication process holds the well-known advantages of simple process, low-cost, and batch fabrication, however, the tradeoff is the poor electrical conductivities of the material and the thin oxide layer that grows on the surface which limits the current. For our application in energy harvesting fields, it may decrease the energy transfer efficiency because of the energy consumption caused by the poor electrical conductivity if a highly doped substrate is not used. Micro-discharges could also vaporize or sputter the electrode material after a long-term running at high current [57], which increases the pressure in the gap spacing if device is packaged and changes the gap itself thus decreases the breakdown voltage. Employing tough metal materials will help to improve the emission efficiency and stability. For example, tungsten coating on silicon-based gated emitters have been used[76][77]. However, we have to notice that such kind of tungsten coating or metal deposition will decrease the turn-on/breakdown voltage [57], which is on the contrary with our purpose of increasing the operation voltage as high as possible. Therefore, the system can benefit from the good conductivity and low leakage of the metal materials, only on the basis of proper design for not decreasing the breakdown voltage.”

[57]. T. Ono, D. Y. Sim, M. Esashi, Micro-discharge and electric breakdown in a micro-gap. *Journal of Micromechanics and Microengineering*. 10, 445 (2000).

[59]. C. H. Chen, J. A. Yeh, P. J. Wang, Electrical breakdown phenomena for devices with micron separations. *Journal of Micromechanics and Microengineering*. 16, 1366 (2006).

[76]. L. Chen, M. M. El-Gomati, Fabrication of tungsten-coated silicon-based gated emitters. *Journal of Vacuum Science & Technology B: Microelectronics and Nanometer Structures Processing, Measurement, and Phenomena*. 17, 638-41 (1999).

[77]. T. Ramsvik, S. Calatroni, A. Reginelli, M. Taborelli, Influence of ambient gases on the dc saturated breakdown field of molybdenum, tungsten, and copper during intense breakdown conditioning. *Physical Review Special Topics-Accelerators and Beams*. 4, 042001 (2007).

Q8- Based on your data, the choice of $g_0 = 9 \mu\text{m}$ seems to be a fortunate coincidence; what would it take to optimize the design? In other words, maybe there is a better value for g_0 , with even better performance. What would it take to find it?

Response: We thank the reviewer for this good question. Actually, with unstable charge pump like our circuit, there no existing optimized design because we always have better performance by approaching a higher $V_{C_{buf}}$ and a narrower hysteresis loop. However, $V_{C_{buf}}$ and the width of the hysteresis loop are limited by three factors: (1) the inverse breakdown voltage of the diodes (it is $\sim 570\text{V}$ in this paper), which defines the upper limit voltage of $V_{C_{buf}}$; (2) the ON voltage of the switch (from $\sim 300\text{V}$ to $\sim 450\text{V}$ in this paper), which defines the working voltage of $V_{C_{buf}}$; and (3) the passing current when the switch is ON, which defines the charge releasing time, i.e. the width of the hysteresis loop.

We added some descriptions in the discussion part (p.14, marked in red):

“However, there is still a large space to further improve the performances of the circuit. For example, we can have more diodes in series to get a higher inverse breakdown voltage, which will increase the upper limited working voltage for the buffer capacitor. According to the Paschen’s law, we can increase the gap to tens of or even one-hundred micrometers to approach a $\sim\text{kV}$ ON voltage. At the same time, the number of pairs of tips can be reduced to narrow the hysteresis. However, keeping a proper redundancy of the tips ensures the robustness and sustainability of the switch.”

Q9- Page 11, you mention that 80 pairs of tips are working in the fixed-fixed switch. I respectfully disagree. The microplasma is a non-linear phenomenon, it is very sensitive to the tip radii (tip electric fields depend on their tip radius), my guess is that only a few of them are working during the discharge due to the tip radii spread (it is unavoidable when you make arrays of anything, and when one makes arrays of very small features, the spread tends to have long tails).

Response: We agree that not all of the 80 pairs of tips work at the same discharge cycle. Although it cannot be precisely localized, we can still find proof from the Supplementary Video 1: the appearance of the flare is not always fully distributed over the 80 tips when the switch is ON and there is a slight difference in the flash positions for each time. Another proof is that the ON/OFF voltage at each switch operation cycle is not exactly the same, seen from Fig. 3f (previous Fig. 3c).

We added the discussions as follows (p. 12, marked in red):

“The narrower hysteresis of the movable switch ($g_0 = 9\mu\text{m}$) compared to the fixed switch can be explained by the fact that only one pair of triangular tips is working for the movable switch, whereas multi pairs of tips are working in the fixed switch. But it has to be noted that not all of the 80 pairs of tips are working at the same time for a specific discharge cycle. Indeed, we can see from the Supplementary Video 1 that the flare did not appear all of the positions of the tips and there is a slight difference time to time. The disparity of the working tips results in the slight ON/OFF voltage variations as shown in Fig. 3e.”

Figure 3 | Designed MEMS fixed plasma switch and its electrical characteristics. (a) Schematic and SEM image of the MEMS fixed plasma switch with an array of triangle tips as the discharging electrodes. (b) The simulated electric field of three tips with voltages of 200V and 400V between the anode and the cathode. (c) Photo of the MEMS switch in the OFF state. (d) Photo of the MEMS switch in the ON state. The figure between (c) and (d) shows the MEMS chip and the setup to capture the plasma discharge including a microscope and camera. (e) The output voltage across C_{buf} and C_{store} when using the fixed sharp switch (a) in the 2-stage circuit without any regulator and load. (f) A detail of the output voltage in (e) between 30s and 34s. (g) Comparisons of the voltages across C_{store} with different loads.

Q10- Talking about the multi-tip switch, why do you have so many tips? are you concerned about lifetime (so as soon as the sharper tips get damaged, the duller tips start working)? please comment

Response: The first point we concerned about when designing the device was that multi-tip design would generate a higher total switch-ON passing current compared to the single-tip design, while keeping a relatively narrow-hysteresis. This concern has been verified by comparing Fig. 3e and 4f. On the other hand, we agree with the reviewer that the multi-tip design will improve the reliability, as said, if one tip gets damaged, the other tips can still work. But the lifetime is not the major concern in the design. Actually, for our experiments, even with only one pair of tips as the movable switch, it can work for a long time. Furthermore, the multi-tip design will decrease the ON current per emitter [73], thus weaken the risk of physical damage.

We added the following comments (in page 12, paragraph 1, 3, marked in red):

“In the design, we expect the multi-tip device (fixed switch) generates a higher ON current compared to the single-tip device (movable switch), while still keeping a narrow hysteresis, which can be actually proved by comparing the amplitude of the hysteresis loop in Fig. 3e and 4f.”

“The other advantage the multi-tip design brought in compared to the single-tip design is the improved robustness: some tips can still work if one or several tips get damaged for any reason, and the ON current per emitter is lower in comparison with the single-tip design [73]”

[73]. S. A. Guerrero, A. I. Akinwande, Nanofabrication of arrays of silicon field emitters with vertical silicon nanowire current limiters and self-aligned gates. *Nanotechnology*. 27, 295302 (2016).

Q11- Supplementary video 2. Maybe is my browser, but your video is upside down! can you please check you uploaded the video in the right orientation?

Response: The supplementary video was corrected.

Q12- Thinking some more about your multi-tip device, the authors should point out that you might need current regulators in each emitter to uniformize the current in the device, to protect the tips from burning/damaging, to efficiently use the array of tips. In a nutshell, you can take care of the tip radii spread if you integrate negative feedback, i.e., having some ballast in series with each tip, so there is a maximum current per emitter, so the sharper tips can work at the same time the duller tips work. An ideal ballast has high impedance and high saturation current, e.g. an ungated field effect transistor (FET) operating in saturation, which is not hard to achieve given the voltage involved (you can get the FETs to saturate with volts). Given that you are making your structures in silicon, you can easily make long and narrow fingers ending in tips, which would make the fingers act as ungated FETs monolithically integrated to the sharp tips. Researchers have reported a similar idea for field emission of electrons:

-L. F. Velasquez-Garcia et al. 2011 "Uniform High-Current Cathodes Using Massive Arrays of Si Field Emitters Individually Controlled by Vertical Si Ungated FETs—Part 1: Device Design and Simulation" IEEE Transactions in Electron Devices, vol. 58, No. 6, pp. 1775 - 1782.

please add such discussion. Again, this is a significant issue for your approach to be adopted mainstream.

Response: We appreciate the reviewer for this very interesting suggestion. We added the following paragraph in the discussion part (p.15, marked in red):

“To definitely ensure that no reliability issues would occur after a long period of actuation, current regulators might be added to protect the silicon tips of the electrodes. This can be obtained by integrating a negative current feedback, i.e. having some current limiters in series with the silicon tips, as proposed in [78], where thin and long silicon tips implemented ungated MOS transistors providing a current limitation. ”.

[78]. L. F. Velasquez-Garcia, S. A. Guerrero, Y. Niu, A. I. Akinwande, Uniform high-current cathodes using massive arrays of Si field emitters individually controlled by vertical Si ungated FETs—Part 2: Device fabrication and characterization. IEEE Transactions on Electron Devices 58, 1783-91 (2011).

REVIEWERS' COMMENTS:

Reviewer #2 (Remarks to the Author):

You did a very good job at addressing all the feedback from the Reviewers. The changes motivated by your response greatly improved the quality of your manuscript. In my opinion, this paper is now excellent and should be accepted for publication in its current form. I hope you are doing well in the midst of the pandemic. Good luck in your research!